# Impact of Dietary Supplementation with *Pogostemon cablin* Essential Oil on the Rumen Fermentation and Rumen Health in Heat-Stressed Beef Cattle

**DOI:** 10.3390/ani15213123

**Published:** 2025-10-28

**Authors:** Chuntao Nie, Xilong Wu, Xianglong Shang, Huan Chen, Lin Li, Lanjiao Xu, Xiaozhen Song

**Affiliations:** Jiangxi Province Key Laboratory of Animal Nutrition, Engineering Research Center of Feed Development, Jiangxi Agricultural University, Nanchang 330045, China; 16668228134@163.com (C.N.); westlife2012623@163.com (X.W.); sxl221011@163.com (X.S.); 15797863693@163.com (H.C.); 13657919889@163.com (L.L.)

**Keywords:** *Pogostemon cablin* essential oil, heat stress, beef cattle, rumen health

## Abstract

**Simple Summary:**

The rumen is a vital organ in ruminants. Existing studies have shown that heat stress can negatively affect the rumen and compromise the integrity of the ruminal epithelial tissue. This study investigated whether dietary supplementation with *Pogostemon cablin* essential oil (PEO) improves rumen health in heat-stressed beef cattle. The results showed that PEO enhanced nutrient digestibility, increased antioxidant enzyme activity, promoted the repair of damaged rumen epithelium, and exerted a protective effect on rumen function in heat-stressed cattle. These findings underscore the potential of PEO as a novel feed additive for alleviating heat stress in beef cattle.

**Abstract:**

This study aimed to investigate the effects of *Pogostemon cablin* essential oil (PEO) on rumen development in heat-stressed beef cattle. Eighteen male Jingjiang cattle were randomly assigned to two groups and fed a diet containing PEO at 0 mg/kg (Control) and 50 mg/kg in the feed concentrate (n = 9 per group). The rumen fluid samples had lower ammonia nitrogen and higher cellulase activity, propionate and total volatile fatty acids concentrations in the 50 mg/kg PEO group. Compared with the control group, 50 mg/kg dietary supplementation with PEO increased crude protein and neutral detergent fiber digestibility. Additionally, the ruminal tissue papilla height, the papilla surface area, and the activities of glutathione peroxidase, total superoxide dismutase, and total antioxidant capacity were also higher, while the malondialdehyde content was lower for the heat-stressed cattle in the 50 mg/kg PEO group. Furthermore, PEO increased the average optical density values and mRNA expression of zonula occludens-1 (ZO-1) and occludin (*p* < 0.05). Transcriptomics analysis of the rumen epithelium showed that PEO upregulated the expression levels of genes related to tight junction proteins and the DNA replication/repair pathways, while it downregulated pro-apoptotic genes. In summary, dietary PEO supplementation improved nutrient digestibility, enhanced rumen antioxidant capacity, and promoted the repair of damaged rumen epithelium in heat-stressed cattle, indicating that PEO exerts a prominent protective effect on rumen function.

## 1. Introduction

Extremely high temperatures have become more common in recent years due to global warming. Livestock are exposed to multiple stressors caused by climate change, which may exacerbate thermal stress and thereby reduce productivity and profitability through declines in feed efficiency, growth rates, and reproduction rates [1]. Heat stress can impair optimal production through multiple mechanisms, such as negatively affecting key productive parameters including milk yield, growth, and reproduction [2]. The rumen, the primary forestomach of ruminants, serves as a vital digestive organ that plays a critical role in nutrient metabolism. However, heat stress is known to adversely affect rumen functionality. During heat stress, blood flow to the rumen epithelium is reduced, and the acid–base homeostasis is altered [3]. The end result is ruminal acidosis because of compromised buffering capacity [4], which may impair rumen functionality and damage the rumen epithelium’s integrity. In addition, heat stress causes the imbalance of the rumen microbial community and affects rumen peristalsis [5]. These heat stress-induced alterations in rumen motility and microbiota negatively impact feed digestibility and fermentation efficiency [6]. Thus, some feed additives have been investigated for their action on the animal physiology to cope with heat stress in livestock.

*Pogostemon cablin* Benth., commonly referred to as patchouli, serves as a key ingredient in various renowned traditional Chinese medicinal formulations. Notable examples include its application in Huoxiang Zhengqi Koufuye (oral liquid preparations) and Baoji Pian (tablet formulations). At present, the analysis of and research on the chemical composition of patchouli mainly focus on the volatile oil, which is also known as *Pogostemon cablin* essential oil (PEO) [7]. The antioxidant and anti-inflammatory activities of PEO have been reported [8]. For over 2000 years, *Pogostemon cablin* has been renowned for its “dispelling heat and dampness” and is recognized as a cure for heat stress. Currently, the Chinese patent drug Huoxiang Zhengqi San, which contains patchouli as its primary component, is widely used in clinical settings for the treatment of heat stroke [9]. In animal husbandry, El-Zaiat and Abdalla [10] found that PEO could modulate the ruminal fermentation pattern and function as a ruminal modifier due to its abatement potential on methanogenesis. Furthermore, *Pogostemon cablin* finds application in traditional Chinese medicine (TCM) to alleviate gastrointestinal disorders [11]. Liu et al. [12] showed that the Huoxiang Zhengqi Oral Liquid enhances the ultrastructure of intestinal epithelial cells, upregulates the expression of zonula occludens-1 (ZO-1) and occludin, repairs the colonic epithelial barrier, and reduces intestinal permeability. These results suggested that PEO exerts modulatory effects on ruminal fermentation and stabilizes the rumen environment in ruminants. However, limited studies have focused on its effects on heat stress in animals, and the specific regulatory mechanisms and action pathways remain unclear. In our previous study, our team systematically evaluated the effects of dietary supplementation with PEO at four different concentrations (0, 50, 100, and 150 mg/kg) on heat-stressed beef cattle. The results indicated that a dosage of 50 mg/kg yielded the most significant improvements in growth performance and immune function. Furthermore, PEO at 50 mg/kg was found to help restore the rumen epithelial barrier [13]. Therefore, the current study was designed to further investigate the effects and mechanisms of this pre-identified optimal dose. The study aimed to investigate the effects of PEO on nutrient digestibility, rumen antioxidant capacity, and rumen epithelial integrity. Additionally, we sought to investigate and elucidate on the mechanisms by which PEO alleviates heat stress in beef cattle and identify the pathways through which patchouli oil protects rumen epithelial health.

## 2. Materials and Methods

### 2.1. Animals, Experimental Design, and Diets

The study was conducted in the high-temperature and high-humidity environment during summer (from 11 July to 8 September). The environmental conditions throughout the experimental period, as recorded in our parallel trial at the same site [13], averaged 30.54 °C in temperature and 79.22% in humidity, resulting in a mean temperature–humidity index (THI) of 83.4. PEO (purity of 98%, contains 27.38% patchouliol, 19.25% α-patchoulene, 15.23% α-guaiene, etc.) was obtained from Jiangxi Zhongcheng Pharmaceutical Co., Ltd., Jiangxi province (Nanchang, China). This essential oil was then converted into a stable powder form by Hangzhou King Techina Feed Co., Ltd. (Hangzhou, China). using β-cyclodextrin inclusion technology. The final product contained 60% (*w*/*w*) Pogostemon cablin essential oil with the remaining 40% consisting of the β-cyclodextrin carrier.

Eighteen healthy male Jingjiang cattle, 18 months old and 330 ± 20 kg of body weight, were selected using a single-factor completely randomized design. All cattle were housed individually to eliminate social hierarchy or group dynamics as confounding factors. Pre-feeding was conducted for 10 days, which was followed by an experimental period of 60 days. A total of 18 cattle were blocked by their initial body weight and then randomly allocated within each block to one of two dietary treatments: a control group fed a basal total mixed ration (TMR) or a treatment group fed the basal TMR supplemented with 50 mg/kg PEO (n = 9 per group). The PEO product was first premixed into the concentrate portion of the diet. A complete TMR was then prepared in batches by thoroughly mixing this PEO-fortified concentrate with wet brewers grains and chopped rice straw. Throughout the trial, all cattle were fed this TMR ad libitum twice daily (at 06:00 and 17:00). The amount of feed offered was adjusted daily to ensure approximately 10% orts (feed refusals). Table 1 shows the composition and nutrient levels of the basal diet. NE_mf_ was calculated according to the Chinese Feeding Standard of Beef Cattle (NY/T 815-2004) [14].

### 2.2. Feed and Fecal Samples Collection and Analysis

Over the final five consecutive days of the experiment, fecal samples were obtained from each animal via rectal sampling at 07:00 and 18:00 h daily. During this same collection period, samples of feed ingredients were also gathered. Daily fecal samples were collected and stored separately for each individual animal. For subsequent analysis, a representative subsample equivalent to 3% of the daily wet weight was taken from each animal’s daily collection and split into two portions: one portion was mixed with 10% (*v*/*v*) sulfuric acid to fix nitrogen and stored at −20 °C for the determination of crude protein via the Kjeldahl method. The other portion was kept without acid addition, stored at −20 °C, and used for the analysis of other conventional nutrients, including dry matter, ash, and acid-insoluble ash (AIA). Diet and fecal samples were oven-dried at 65 °C for 72 h, ground using a shredder, and passed through a 40-mesh sieve (aperture size of 0.425 mm). Neutral detergent fiber (NDF), acid detergent fiber (ADF), and acid-insoluble ash (AIA) in feed and fecal samples were analyzed using the method of Van Soest et al. [15]. Concurrently, the ether extract (EE), crude protein (CP), calcium and phosphorus in feed and fecal samples were conducted following standardized procedures of AOAC [16]. The apparent digestibility of nutrients was determined by the acid-insoluble ash (AIA) method, employing AIA concentration as an internal indigestible index [17].

### 2.3. Rumen Fluid Samples Collection and Analysis

Following our team’s dose-finding study that evaluated growth status and serum parameters [13], eight cattle from the control and 50 mg/kg PEO groups (four individuals per therapy with medium body weight) were identified for the sampling of rumen fluid and epithelial tissue for subsequent analysis. At the end of the feeding trial, after a 16 h fasting period (with free access to water) following the final feeding, cattle were transported to a slaughterhouse approximately 500 m away and humanely euthanized following institutional animal care protocols. At slaughter, rumen fluid was collected immediately from the rumen. The rumen fluid was filtered through four layers of surgical gauze. Ruminal pH was measured with a pH meter immediately after sampling (model PHS-3C, Shanghai Precision Instrument Co., Shanghai, China). Then, it was dispensed into three 15 mL centrifuge tubes. The rumen fluid samples in centrifuge tubes were flash-frozen in liquid nitrogen and stored at −20 °C for the analysis of rumen fermentation parameters and digestive enzyme activity.

Digestive enzyme activity (including cellulase [EC 3.2.1.4], α-amylase [EC 3.2.1.1], and lipase [EC 3.1.1.3]) in rumen fluid samples was analyzed using commercial assay kits (Nanjing Jiancheng Bioengineering Institute, Nanjing, China). The specific product model numbers and sample pretreatment protocols were as follows. Lipase (Cat. No. A054-1-1): Rumen fluid was centrifuged at 2500× *g* for 10 min, and the supernatant was collected for analysis. Cellulase (Cat. No. A138): Rumen fluid was centrifuged at 4000× *g* for 10 min at room temperature. A portion of the resulting supernatant was taken for immediate analysis, while another portion was placed in a boiling water bath for 5 min and then cooled to serve as a boiled supernatant control. α-Amylase (Cat. No. ADS-W-DF002): Rumen fluid was centrifuged at 12,000× *g* for 10 min at 4 °C, and the supernatant was placed on ice for subsequent measurement. Each sample was measured in 3 replicates with a spectrophotometer. The concentrations of volatile fatty acids (VFAs) were quantified by a gas chromatograph (Agilent Technologies 7820A, Santa Clara, CA, USA) fitted with a free fatty acid phase capillary column (30 m × 0.25 mm × 0.33 μm, Lanzhou Atech, Lanzhou, China). Prior to analysis, rumen fluid samples were prepared as follows: a 1.5 mL aliquot was centrifuged at 10,000× *g* for 15 min at 4 °C. Then, 1.25 mL of the supernatant was mixed with 0.25 mL of ice-cold 25% (*w*/*v*) metaphosphoric acid. The mixture was centrifuged again under the same conditions (10,000× *g*, 15 min, 4 °C). The resulting supernatant was filtered through a 0.45 μm aqueous phase membrane before injection into the chromatograph. The concentration of ammonia nitrogen (NH_3_-N) was quantified with the phenol–hypochlorite colorimetric method. For this analysis, a 0.5 mL aliquot of rumen fluid was centrifuged at 4000× *g* for 10 min. A 100 μL aliquot of the supernatant was then diluted with 300 μL of distilled water, which was followed by the addition of 1.6 mL of 0.2 mol/L HCl to make a final volume of 2 mL for measurement. The ruminal microbial protein (MCP) content was determined via a colorimetric assay using a commercial enzyme-linked detection system (Suzhou Geruisi Biotechnology Co., Ltd., Suzhou, China; Cat. No. G0418W) following the manufacturer’s spectrophotometric protocol. Briefly, rumen fluid samples were thawed and vortexed for 45–60 s. A 1.5 mL aliquot was centrifuged at 700× *g* for 5 min. Then, 1 mL of the resulting supernatant was transferred to a new tube and centrifuged at 12,000 rpm for 30 min at 4 °C. The liquid supernatant was used for the subsequent analysis. All samples were measured in triplicate.

### 2.4. Rumen Epithelial Tissue Sample Collection and Preparation

During slaughter, the rumen was separated, and ruminal contents were quickly taken out. A 2 cm × 2 cm piece of rumen wall was cut from the rumen dorsal sac with surgical scissors and rinsed with cold sterile PBS (pH 7.4). The rumen tissue samples were divided into 4 parts, 2 of which were cut into small pieces (about 1 g), rapidly frozen in liquid nitrogen, and transported to the laboratory for storage at −80 °C pending quantitative real-time PCR and antioxidant capacity assays. The other two parts were cut into 1 cm × 1 cm and 0.5 cm × 0.5 cm pieces, which were then preserved in 10% formalin solution and 4% paraformaldehyde (PFA) for ruminal papillae measurement and immunofluorescent staining, respectively. To ensure that whole papillae were obtained, the cut was made below the base of the papillae.

### 2.5. Morphological Characteristics of Rumen Papillae and Immunofluorescent Staining

Tissue samples in 10% formalin solution were used for determining macroscopic papillary morphology, including the length, width and density. A total of 30 intact and well-oriented papillae per animal were measured for length and width according to the method of Van Niekerk [18]. The rumen epithelial tissues fixed in 4% PFA were dehydrated in an ethanol gradient (70%, 75%, 85%, 90%, 95%, and 100%) and embedded in paraffin. Following dehydration and embedding, the paraffin sections were dewaxed and blocked with 5% bovine serum albumin for 60 min. Subsequently, primary antibodies (diluted 1:250 in PBS) were applied to sections and incubated overnight at 4 °C. The antibodies included the following: Rabbit anti-Claudin-1 polyclonal antibody, Rabbit anti-Occludin polyclonal antibody, and Rabbit anti-Zonula Occludens-1 polyclonal antibody (Servicebio Technology Co., Ltd., Wuhan, China). After incubation of the primary antibodies, slices were incubated with the secondary antibodies (diluted 1:500 in PBS) at room temperature for 60 min. Secondary antibodies included TSA-FITC-conjugated goat anti-rabbit IgG and TSA-CY3-conjugated goat anti-rabbit IgG. Sections were further incubated under light-protected conditions for 50 min to minimize photobleaching. After final washing, sections were imaged using a Nikon A1R confocal laser-scanning microscope (Nikon Corporation, Tokyo, Japan) with excitation wavelengths of 405 nm and 488 nm. Quantitative analysis of claudin-1, ZO-1, and occludin expression was performed by measuring average optical density with Image-Pro Plus 6.0 software (Media Cybernetics, Rockville, MD, USA).

### 2.6. Determination of Antioxidant Index and Real-Time PCR

Approximately 0.5 g of frozen rumen tissue was placed into a cryovial containing sterile grinding beads. Physiological saline was added at a 1:9 (*w*/*v*) tissue-to-saline ratio and then ground using a grinder. After centrifugation, we took the supernatant and aliquoted it into centrifuge tubes. The malondialdehyde (MDA; Cat. No. A003-1) concentration, total superoxide dismutase (T-SOD; Cat. No. A001-3) activity, glutathione peroxidase (GSH-Px; Cat. No. A061-1) activity, and total antioxidant capacity (T-AOC; Cat. No. A015-3-1) in the supernatant were quantified using a commercial detection kit (Nanjing Jiancheng Bioengineering Institute, Nanjing, China) according to the manufacturer’s protocol.

Rumen epithelial RNA isolation was conducted using TRIzolTM reagent (TransGen Biotech, Beijing, China). RNA purity and integrity were assessed with a NanoDrop™ micro-spectrophotometer (Bio-Rad, Hercules, CA, USA). The 260/280 nm absorbance ratios of all samples ranged from 1.8 to 2.0, indicating high RNA purity. The transcribed RNA was reverse-transcribed according to the TransScript All-in-one First-Strand cDNA Synthesis SuperMix kit (Quanshi Jin, Beijing, China) to become more stable cDNA. Primer sequences for claudin-1, occludin, ZO-1, and the reference gene β-actin were designed with PrimerSelect software (v7.1, DNASTAR, Madison, WI, USA) based on NCBI GenBank sequences and synthesized by BGI Genomics (Wuhan, China). The primer sequences are shown in Table 2. QPCR was performed on synthesized cDNA according to the manufacturer’s protocol using the PerfectStart^®^ Green qPCR SuperMix (TransGen Biotech, Beijing, China). Amplification conditions were as follows: initial denaturation at 95 °C for 30 s, then denatured at 95 °C for 10 s, annealed at 60 °C for 30 s, and extended at 72 °C for 30 s for a total of 40 cycles. All measurements are in triplicate. Using β-actin as a reference gene, the relative expression levels of these genes were calculated by the 2^−ΔΔCT^ method.

### 2.7. Identification of Differentially Expressed Genes and Functional Annotation of the Rumen Epithelial Tissue

The total RNA was extracted from the rumen epithelium. After confirming RNA quality, high-throughput transcriptomic sequencing was performed by Shanghai Meiji Biomedical Technology Co., Ltd. (Shanghai, China). For transcriptomic analysis, RNA samples from three heat-stressed beef cattle were randomly selected from both the 50 mg/kg PEO treatment group and the control group for library preparation and sequencing. Briefly, the raw data obtained are processed to remove adapter contamination and low-quality readings. Through the Ensembl database (http://asia.ensembl.org/Bos_taurus/Info/Index, accessed on 15 November 2022), clean reads and the cattle reference genome (Bos Taurus ARS–UCD1.2), we transcribed the assembly and quantitative expression. Gene expression levels were normalized as fragments per kilobase of exon per million mapped fragments (FPKM). Differentially expressed genes (DEGs) were identified by the DESeq2 method with the following threshold values. The comparison results were screened with the thresholds of |log2FC| ≥ 0.585 and *p*-value < 0.05. Only genes whose expression levels were above the detection threshold (FPKM > 0.1 for each sample) were subjected to further analysis. Then, we conducted quantitative analysis on each read of the six sequenced samples (B1, B2, B3, C1, C2, C3). To characterize the biological functions of DEGs, Gene Ontology (GO) and Kyoto Encyclopedia of Genes and Genomes (KEGG) pathway enrichment analyses were performed using the clusterProfiler R package (v4.0.5). The pathway with a correction of *p* < 0.05 was considered to be significantly enriched. The pathways that meet the conditions of *p* < 0.05 were defined as significant enrichment pathways. Additionally, all DEGs were functionally annotated against the KEGG database to explore pathway associations and genomic network interactions. To validate transcriptomic findings, reverse transcription quantitative PCR was performed on rumen epithelial tissues from six heat-stressed cattle. Primer sequences are provided in Table 2.

### 2.8. Statistical Analysis

Data are expressed as the mean and standard error of the means (SEM). The effects of diet, environmental conditions (summer climate with high temperature and humidity) and time were considered as fixed factors, and the effect of cattle was considered as a random factor. Data generated in the present study were analyzed with independent sample *t*-tests using IBM SPSS Statistics 26.0 (IBM Corp., Armonk, NY, USA). The normality of data was assessed using the Shapiro–Wilk test. For non-normally distributed data, the Mann–Whitney U test was applied. The homogeneity of variance was evaluated by Levene’s test with Welch’s *t*-test employed when variances were unequal. Statistical significance was set at *p* < 0.05 with trends defined as 0.05 ≤ *p* < 0.10.

## 3. Results

### 3.1. Apparent Nutrient Digestibility

Table 3 shows that supplementing with 50 mg/kg PEO increased the CP (*p* < 0.001) and NDF (*p* = 0.026) digestibility, and it tended to increase the ADF (*p* = 0.074) digestibility. Nevertheless, the EE, calcium and phosphorus digestibility had no significant (*p* > 0.05) effect in the 50 mg/kg PEO group compared to those in the control group.

### 3.2. Morphological Characteristics of Rumen Papillae

Figure 1A,B illustrate the different development of papillae from beef cattle fed up to 0 mg/kg and 50 mg/kg of PEO during heat stress. As shown in Figure 1C–F, 50 mg/kg dietary supplementation with PEO increased the papilla height (*p* = 0.009) and papilla surface area (*p* = 0.007) compared with the control group. However, the papillae density did not significant differences between the control group and the 50 mg/kg PEO group (*p* > 0.05).

### 3.3. Rumen Antioxidant Parameters of Ruminal Tissues

As shown in Table 4, compared with the control group, the GSH-Px (*p* = 0.015), T-SOD (*p* = 0.001) activity and T-AOC (*p* < 0.001) in the 50 mg/kg group were increased, while 50 mg/kg PEO decreased the MDA content in rumen tissue (*p* < 0.001).

### 3.4. Rumen Fermentation Parameters and Digestive Enzyme Activities in Rumen Fluid

In Table 5, 50 mg/kg dietary supplementation with PEO increased the propionate (*p* = 0.027) and VFA (*p* = 0.041) contents, while it decreased the NH_3_-N concentration (*p* = 0.008) compared with the control group. However, the two groups exhibited no significant differences with respect to pH, concentrations of MCP, propionate, butyrate, and the acetate-to-propionate ratio (A:P) (*p* > 0.05).

In the rumen, PEO supplementation enhanced the activity of cellulase (*p* = 0.031) when compared to those of the control group (Table 5), while the activity of α-amylase and lipase was not different among the two groups (*p* > 0.05).

### 3.5. The Distribution and Expression of Tight Junction Protein

The results are summarized in Figure 2. Immunofluorescence staining of ZO-1 and occludin in the control group exhibited irregular, diffuse distribution patterns with discontinuous punctate signals localized at cell–cell junctions (Figure 2A–C). In contrast, the PEO group (50 mg/kg) showed a continuous linear localization of claudin-1, ZO-1, and occludin along epithelial surfaces (Figure 2D–F). In addition, as shown in Figure 2G, dietary supplementation with 50 mg/kg PEO increased the average optical density values of ZO-1 (*p* < 0.01) and occludin (*p* < 0.05) in the rumen epithelium. In Figure 2H, 50 mg/kg dietary supplementation with PEO increased the mRNA expression of occludin (*p* = 0.016) and ZO-1 (*p* = 0.015) in the ruminal epithelium compared with the control group.

### 3.6. Sequencing Data Quality Assessment

Prior to sequencing data analysis, data quality and reliability were rigorously assessed to generate high-quality datasets. As detailed in Table A1, each sample yielded ≥6.87 Gb of filtered reads, accounting for 97.62% of raw data. Sequencing quality metrics—Q20 and Q30—reached 98.16% and 94.62%, respectively, confirming exceptional data integrity. Quality-controlled reads were aligned to the bovine reference genome (Bos taurus ARS-UCD1.2) using HISAT2 (v2.2.1). Alignment rates ranged from 95.34% to 96.58% across samples, validating both the reference genome suitability and data robustness for downstream transcriptomic analyses.

### 3.7. Identification of Differentially Expressed Genes and Functional Annotation

A total of 20,967 uniquely expressed genes were identified in the rumen epithelial tissues. Comparative analysis between the control and 50 mg/kg PEO groups revealed 884 DEGs, comprising 433 upregulated and 451 downregulated transcripts (*p* < 0.05; Figure 3). As shown in Figure 4, a total of 39 DEGs were annotated in the GO database. In the molecular function ontology, the intramolecular oxidoreductase activity, protein disulfide isomerase activity, and mechanosensitive channel activity were the most abundant terms. In the cellular component ontology, the catalytic complex, transferase complex, and protein-containing complex were the most abundant terms. In the biological process ontology, the response to chemical, transmembrane transport, and actively regulated cell aging were the most abundant terms. Multiple significantly enriched pathways are listed in Figure 5 and Table 6, which can be roughly classified into several major categories such as metabolic pathways, signaling pathways, and disease-related pathways. Notably, critical pathways for the rumen barrier function—tight junction, apoptosis, and phagosome—were significantly enriched (*p* < 0.05). The phagosome pathway, closely linked to inflammatory responses, showed prominent DEG enrichment, suggesting PEO’s potential anti-inflammatory effects.

### 3.8. Differentially Expressed Genes Related to the Epithelial Barrier of the Rumen

The DEGs were mapped and functionally annotated via GO and KEGG databases. Analyses identified 10 genes associated with tight junction protein regulation and apoptotic pathways (Table 7). Dietary supplementation with 50 mg/kg PEO increased the expression of four genes that play important roles in the tight junction protein expression pathway (*p* < 0.05), such as Tight Junction Protein 1 (*TJP1*), Tubulin Alpha 1b (*TUBA1B*), Proliferating Cell Nuclear Antigen (*PCNA*), and BCL2 Antagonist/Killer 1 (*BAK1*). Meanwhile, PEO downregulated six genes that mediate apoptosis (*p* < 0.05), such as Apoptotic Protease Activating Factor 1 (*APAF1*), CASP8 and FADD-Like Apoptosis Regulator (*CFLAR*), RAS Guanine Nucleotide-Releasing Factor 2 (*RASGRF2*), Fas Cell Surface Death Receptor (*FAS*), Mitogen-Activated Protein Kinase 7 (*MAP2K7*), and BCL2 Binding Component 3 (*BBC3*). To validate transcriptomic data, six DEGs (*TJP1*, *PCNA*, *RASGRF2*, *FAS*, *APAF1*, *MAP2K7*) with the most significant expression changes were selected for RT-qPCR. Results confirmed concordant expression trends between RNA-seq and RT-qPCR (Figure A1), demonstrating the high reproducibility of the sequencing data.

## 4. Discussion

The THI, which combines the effects of temperature and humidity, has been widely used for decades to assess heat stress in cattle. Hahn et al. [19] defined heat stress severity based on THI ranges: normal, THI < 74; alert, 74 ≤ THI < 79; danger, 79 ≤ THI < 84; and emergency, THI ≥ 84. As confirmed by environmental data from the same trial [13], the temperature–humidity index (THI) persistently exceeded 79 for the duration of the study. This confirms that the test cattle were subjected to continuous heat stress. In TCM, *Pogostemon cablin* has been historically used to treat gastrointestinal disorders, alleviate summer heat syndrome, regulate dampness, and stimulate appetite [20]. However, the anti-heat stress mechanisms of PEO in livestock remain poorly understood, warranting further studies to elucidate the underlying molecular pathways.

Nutrients play a pivotal role in ruminant physiology, providing energy substrates, supporting growth, reproduction and production, while maintaining systemic homeostasis. Improving the digestibility of nutrients is conducive to improving the growth performance of animals. However, under heat stress, nutrient absorption is significantly reduced across species; in cattle, dry matter intake may decline by approximately 30% [21]. Consistent with this, Gao et al. [22] reported that heat stress suppresses feed intake and nutrient digestibility in dairy cows. Notably, in a dose-finding and production performance study conducted by our research team under identical heat stress conditions, supplementation with 50 mg/kg PEO not only showed a strong trend toward increased average daily dry matter intake (ADMI) but also significantly improved the average daily gain (ADG) in beef cattle [13]. The gastrointestinal tract is the main site for nutrient absorption, and its integrity is of crucial importance for the absorption of nutrients. Studies suggest that *Pogostemon cablin* is a medicinal plant for treating gastrointestinal symptoms and has demonstrated protective effects on the gastrointestinal tract [23,24]. Moreover, Huoxiang Zhengqi Oral Liquid has been found to effectively regulate gastrointestinal motility in a two-way manner [11]. By normalizing motility, it likely promotes a more stable ruminal environment, which in turn provides sufficient fermentation time. Thus, these factors may improve the digestion and absorption of nutrients. According to the findings of this study, supplemental 50 mg/kg PEO increased CP and NDF digestibility, and there was also a slight increase in ADF digestibility. These findings corroborate the research findings mentioned above. The enhanced nutrient digestibility observed in the present study by our team provides a physiological explanation for the improved growth performance documented in our companion study [13]. Furthermore, PEO supplementation enhanced the activity of cellulase in the rumen. The rumen is a digestive organ unique to ruminants. The higher digestive enzyme activity in the rumen indicates that the fiber in feed can be degraded more effectively, which is consistent with the NDF digestibility result. The decline in growth performance of ruminants under heat stress is attributed not only to reduced feed intake but also to diminished carbon and energy supply from microbial fermentation [25]. Ruminants lack endogenous fiber-digesting enzymes; instead, they rely on complex microbial communities (consortia) to produce polysaccharide-degrading enzymes with enzymatic activity directly dependent on microbial composition and functionality [26,27]. The enzymatic activity of rumen fluid is a direct manifestation of the microbial metabolic activities. Our results indicated that 50 mg/kg dietary supplementation with PEO enhanced the activity of digestive enzymes in the rumen. Rumen microorganisms digest feed through the action of the enzymes they produce. Based on this, we infer that rumen microbes and enzymes collectively degrade proteins and starches into VFA through synergistic interactions, compensating for energy deficits caused by heat stress-induced hypophagia and improving nutrient utilization efficiency.

The ruminal epithelium, a metabolically active tissue, is prone to oxidative stress due to endogenous reactive oxygen species (ROS) generated during intensive metabolic processes [28]. Heat stress is one of the many factors that causes oxidative stress in vivo. Under heat stress, endogenous free radical production and ROS accumulation increase, while antioxidant capacity declines in animals [29]. The excessive accumulation of free radical can induce unsaturated fatty acids to produce MDA in cells, and MDA promotes cell death by disintegrating cells [30]. In this investigation, PEO treatment enhanced the T-AOC and GSH-Px activity as well as the T-SOD in rumen tissue, concurrently reducing MDA levels. This indicates a potentiation of the local antioxidant defense system within the rumen. The antioxidant capacity of PEO is likely attributable to its primary bioactive component, patchouli alcohol (PA). This is supported by an in vitro study demonstrating that PA alleviates heat shock-induced oxidative stress in intestinal epithelial cells by activating the Nrf2–Keap1 pathway and reducing MDA accumulation [31]. Although direct evidence of PEO’s anti-heat stress effects in ruminants remains limited, its well-documented antioxidant and antimicrobial properties [8,32] support its potential to neutralize free radicals and protect rumen epithelial integrity. Consistent with this mechanism, PEO supplementation increased the ruminal papillary length, width and surface area in heat-stressed cattle. These morphological parameters (papillary length and width) are established biomarkers of rumen mucosal development [33,34]. Previous studies have demonstrated that heat stress reduces feed intake in dairy cows, thereby limiting fermentable substrates available for ruminal microbes and ultimately decreasing ruminal papillary width, perimeter and surface area [35]. These findings suggest that PEO may ameliorate heat stress-induced rumen epithelial damage. Increased papillary dimensions (height and width) directly expand the absorptive surface area of the rumen epithelium [36], which aligns with our observed enhancement in nutrient digestibility. A similar finding showed that dietary supplementation with plant essential oils enhances ruminal digestive efficiency in beef cattle by promoting papillary growth and modulating microbiota–epithelial interactions [37].

In this study, the addition of PEO increased propionate and VFA contents. Elevated ruminal propionate production is strongly correlated with improved growth performance in ruminants [38]. The enhancements in rumen fermentation efficiency and nutrient digestibility observed in this study provide a physiological explanation for the significant improvements in average daily gain reported in our companion dose-finding study using the same PEO supplementation level (50 mg/kg) under heat stress [13]. In ruminants, both acetate butyrate and propionate, the primary energy substrates for ruminants, stimulate ruminal epithelial development. VFA (acetate, propionate, butyrate) modulate key signaling and metabolic pathways—including PPARγ and mTOR signaling pathways and propionate metabolism—via the regulation of genes (e.g., HMGCS2, PPARG, ECHS1, and RNF152) [39]. These pathway modifications drive morphological adaptations in the rumen epithelium, such as increased papillary length and width (as observed herein). Wang et al. [40] further demonstrated that elevated VFA concentrations enhance ruminal maturation during early postnatal stages. This suggests that in addition to enhancing the protective mechanism by increased oxidation resistance, PEO improved epithelial morphology by a trophic effect through increasing propionate concentration, further promoting the improvement of nutrient digestibility. Wang et al. [38] investigated the effects of traditional Chinese medicine compounds containing *Pogostemon cablin* on rumen fermentation via an in vitro gas production technique. Their results demonstrated that traditional Chinese medicine compounds supplementation reduced acetate molar proportion, increased propionate molar proportion, and maintained stable ruminal pH—which are findings consistent with our study. Ruminal NH_3_-N concentration reflects dietary protein degradation efficiency, as NH_3_-N serves as the primary nitrogen source for MCP synthesis by rumen microbiota [41]. In this study, dietary supplementation with PEO reduced the ruminal NH_3_-N concentration in beef cattle. This suggests that PEO may enhance the incorporation of nitrogen into microbial biomass, as NH_3_-N is a primary precursor for microbial protein (MCP) synthesis. Although the observed increase in MCP concentration was not statistically significant, the numerical increase, coupled with the significant reduction in NH_3_-N, indicates a potential for improved nitrogen utilization efficiency that warrants further investigation. The observed reduction in ruminal NH_3_-N concentration with PEO supplementation aligns with the well-documented effects of other plant essential oils. Jahani-Azizabadi et al. [42] reported that essential oils rich in phenolic compounds (e.g., thymol in thyme, cinnamaldehyde in cinnamon) significantly suppress in vitro ruminal NH_3_-N concentration, which is a finding corroborated by subsequent research [43] where dietary supplementation with plant essential oils reduced NH_3_-N levels in goats. Although PEO is characterized by its high content of sesquiterpenoids (e.g., PA) rather than phenolics, it belongs to the same broader class of phytogenic compounds known for their antimicrobial potency. The shared ability of these chemically distinct yet functionally analogous essential oils to lower NH_3_-N suggests a common physiological outcome: the selective modulation of rumen microbiota to inhibit hyper-ammonia-producing microbes. Thus, our results demonstrate that PEO, like other bioactive essential oils, enhances NH_3_-N utilization efficiency during heat stress, thereby optimizing the substrate available for microbial protein synthesis.

The tight junction proteins play a key role in maintaining intercellular adhesion and the physical barrier function of the rumen epithelium [44]. In this study, dietary supplementation with 50 mg/kg PEO significantly upregulated mRNA expression and elevated the average optical density values of ZO-1 and occludin in rumen tissues. Optical density values, quantified via semiquantitative fluorescence signal analysis, directly reflect tight junction protein expression intensity. Immunofluorescence localization revealed distinct tight junction distribution patterns: ZO-1, claudin-1 and occludin exhibited irregular and discontinuous localization in the control group; all proteins displayed continuous, linear distribution along epithelial junctions in the patchouli oil group. As core components of tight junction complexes, ZO-1 and occludin are critical for barrier maintenance. Previous studies confirmed that heat stress compromises rumen epithelial integrity and downregulates tight junction gene expression [45]. Tight junction dysfunction increases intestinal permeability, predisposing to inflammation and gastrointestinal pathologies [46]. Importantly, the protective effect of PEO’s primary bioactive component, PA, on tight junction proteins and intestinal barrier integrity has been well documented. Studies have demonstrated that PA significantly upregulates the expression of ZO-1, ZO-2, claudin-1, and occludin in intestinal epithelial models while concurrently reducing pro-inflammatory cytokines and promoting mucosal repair [47]. Emerging evidence indicates that certain active constituents of Chinese herbal medicines possess prebiotic-like effects that alleviate inflammation and benefit the gut epithelial environment [48,49]. Supporting this, a study demonstrated that PEO and its derived compounds exert significant prebiotic-like effects in the C57BL/6J mouse model [23]. Importantly, in our team’s concurrent dose-finding study conducted under identical heat stress conditions, we observed that PEO supplementation effectively suppressed inflammatory responses in beef cattle [13]. We therefore propose that PEO mitigates heat stress-induced damage by modulating inflammatory processes and enhancing epithelial barrier integrity. Our findings demonstrate that PEO supplementation counteracts heat stress-induced barrier impairment by upregulating ZO-1 and occludin expression, which is likely through the combined effects of suppressed inflammation and enhanced rumen epithelial metabolic activity.

To elucidate the molecular mechanisms by which dietary PEO supplementation enhances rumen barrier function in heat-stressed beef cattle, we performed KEGG pathway enrichment analysis on DEGs. Twenty-four pathways were significantly enriched (*p* < 0.05) with key pathways impacting rumen barrier integrity including: tight junction signaling, phagosome regulation, and apoptosis modulation. Integrated GO and KEGG analyses identified 10 core genes that were involved in the regulation of tight junction proteins and apoptosis. Among the upregulated genes, *BAK1* acts as an antagonist of apoptotic proteins and can inhibit the expression of apoptotic factors [50]; *TJP1* (encoding ZO-1) is a scaffold protein bridging occludin to the actin cytoskeleton [51]; *PCNA* encodes a protein essential for cell replication, which serves as an auxiliary factor for DNA polymerase and plays a critical role in DNA replication and repair [52]. *TUBA1B* encodes α-tubulin, which is a core subunit of microtubules—dynamic cytoskeletal polymers critical for cell division, intracellular transport, and cell shape maintenance. Among the downregulated genes, key apoptotic pathways were identified: exogenous (receptor-mediated) and intrinsic (mitochondrial). The exogenous apoptosis pathway is activated by the *FAS* receptor, and *RASGRF2* enhances the RAS signal. The intrinsic apoptotic pathway begins with the assembly of the cytochrome c complex. This process involves *APAF1* [53]. Additionally, *BBC3* is a pro-apoptotic Bcl-2 family member. Studies have indicated that the activation of the *BBC3* gene in the organism accelerates cell apoptosis [54]. These data indicate that PEO enhances the barrier function of ruminal epithelial cells, promotes cell proliferation and repair, and inhibits apoptosis by upregulating the expression of genes such as *TJP1*, *PCNA*, and *BAK1*. Simultaneously, by downregulating the expression of genes including *APAF1*, *FAS*, and *BBC3*, PEO suppresses apoptosis and the stress response. This coordinated transcriptional reprogramming likely preserves rumen epithelial homeostasis under heat stress, enhancing thermotolerance. Li et al. [55] have shown that acute heat stress can downregulate the expression of genes related to morphogenesis and cytoskeletal structure and upregulate pro-apoptotic genes, thus inhibiting the proliferation of bovine mammary epithelial cells cultured in vitro. This result indirectly supports our conclusion.

## 5. Conclusions

In conclusion, dietary supplementation with 50 mg/kg of *Pogostemon cablin* essential oil (PEO) enhances antioxidant activity, improves nutrient digestibility, promotes ruminal papillae morphology, optimizes rumen fermentation patterns, and stimulates cellulase activity in heat-stressed beef cattle. These findings demonstrate that PEO exerts a significant protective effect on the rumen. Mechanistically, these benefits are achieved by PEO attenuating heat stress-induced damage at the cellular level through suppressing apoptosis pathways, upregulating tight junction proteins to fortify the physical barrier, and enhancing epithelial repair and proliferation. The culmination of these enhanced physiological functions is a significant improvement in the animal’s ability to cope with heat stress (thermotolerance), which directly translates to the previously documented superior growth performance in cattle receiving this supplementation. Therefore, PEO represents an effective strategy to support production outcomes in beef cattle under heat stress.

## Figures and Tables

**Figure 1 animals-15-03123-f001:**
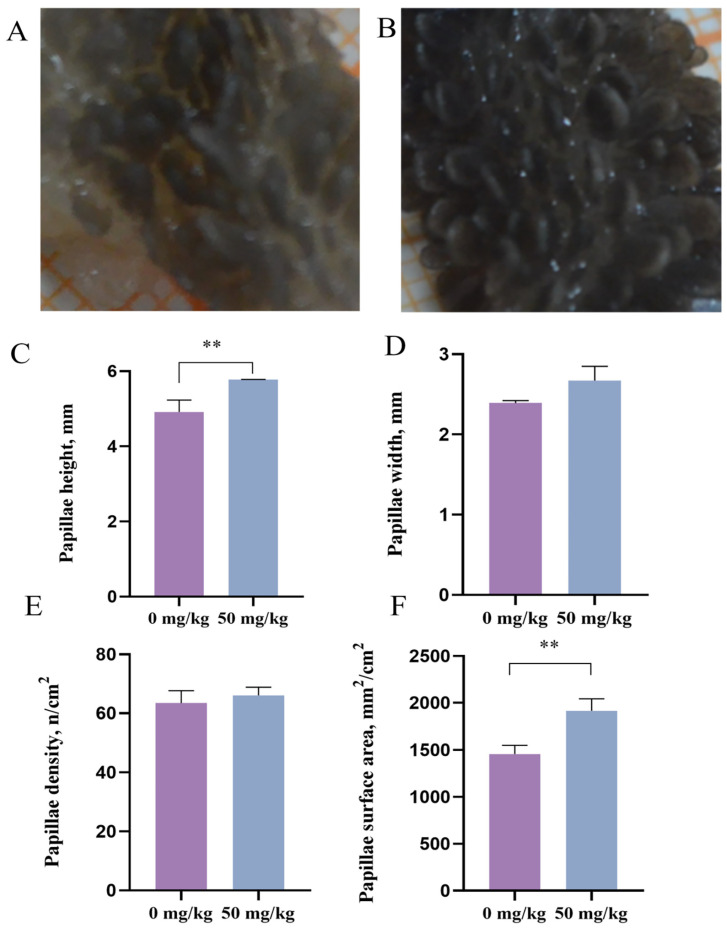
Effects of *Pogostemon cablin* essential oil on rumen papillae morphological characteristics in heat-stressed beef cattle. Representative images of the rumen dorsal sac illustrating different development of papillae from beef cattle fed up to 0 mg/kg (**A**) and 50 mg/kg (**B**) of *Pogostemon cablin* essential oil during heat stress; papillary morphology, including the height (**C**), width (**D**), density (**E**), and papilla surface area (**F**); ** *p* < 0.01.

**Figure 2 animals-15-03123-f002:**
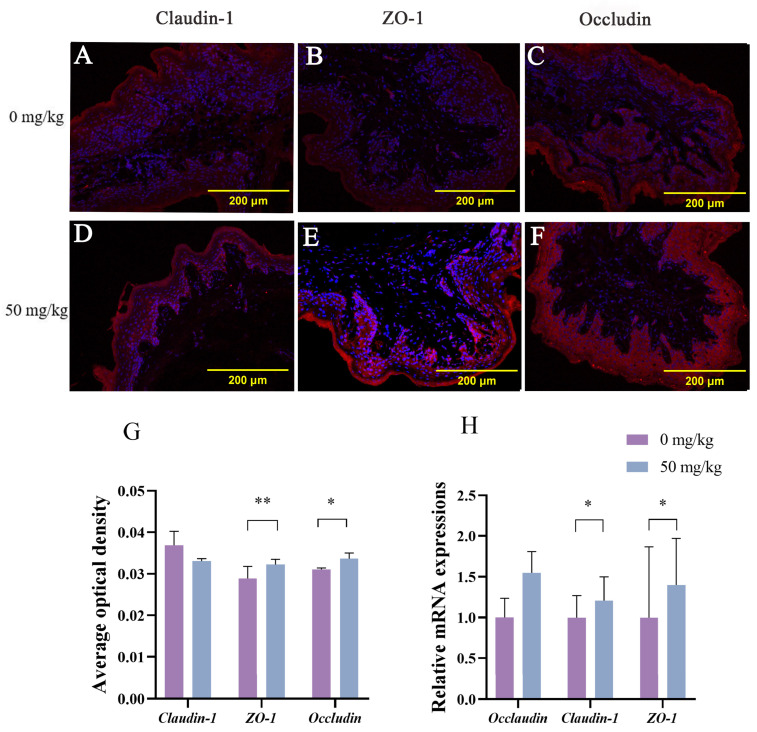
Effects of PEO on distribution and expression of tight junction proteins of heat-stressed cattle. Immunofluorescence localization of claudin-1 (red; (**A**,**D**)), ZO-1 (red; (**B**,**E**)), and occludin (red; (**C**,**F**)) in rumen from the 0 mg/kg group (**A**–**C**) and 50 mg/kg group (**D**–**F**). Average optical density (**G**) and mRNA expression (**H**) of tight junction proteins (claudin-1, ZO-1, and occludin) in rumen epithelium. PEO = *Pogostemon cablin* essential oil. * *p* < 0.05; ** *p* < 0.01.

**Figure 3 animals-15-03123-f003:**
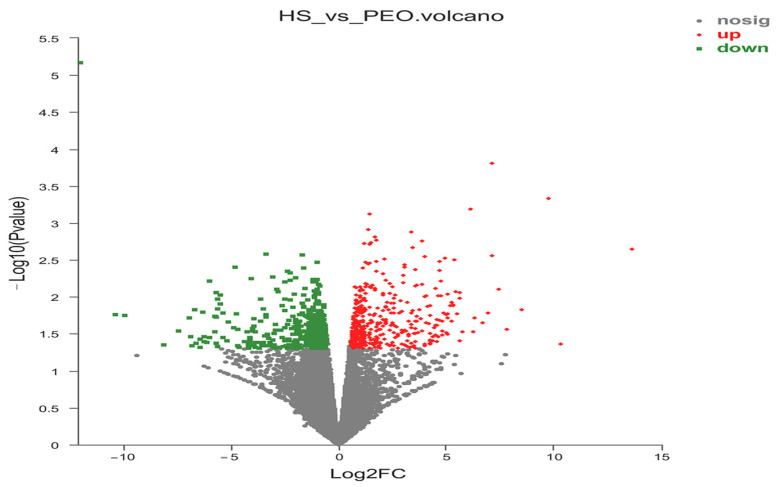
Volcano plot of the differentially expressed genes. The red dots represent the upregulated DEGs, the green dots represent the downregulated DEGs, and the gray dots represent non-DEGs.

**Figure 4 animals-15-03123-f004:**
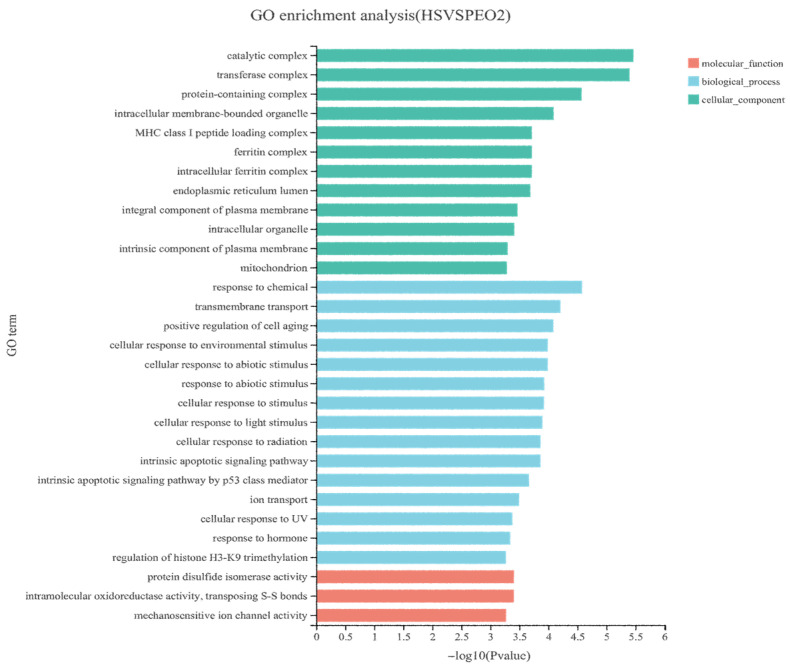
Results of Gene Ontology (GO) analysis in control and the 50 mg/kg PEO group.

**Figure 5 animals-15-03123-f005:**
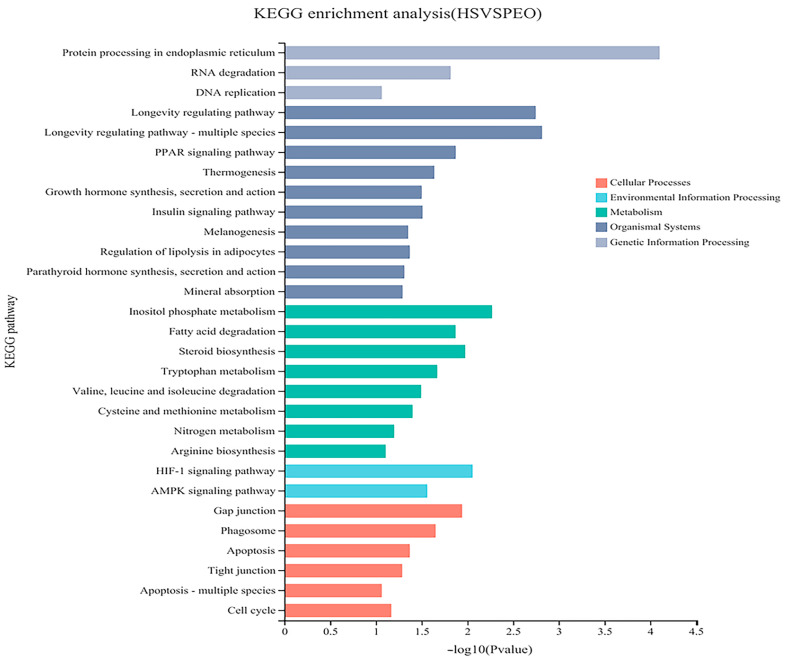
KEGG pathway analysis of relative gene expression in control and 50 mg/kg PEO.

**Table 1 animals-15-03123-t001:** Composition and nutritional levels of diet (% of DM ^1^).

Ingredients	Content	Nutrient Levels	Content
Brewers grains	20.0	DM	90.81
Rice straw	40.0	NE_mf_/(MJ/kg) ^3^	7.25
Corn	21.5	Crude protein	12.69
soybean meal	15.0	Neutral detergent fiber	40.30
NaCl	0.5	Acid detergent fiber	24.89
NaHCO_3_	1.0	Calcium	1.00
Premix ^2^	2.0	Phosphorus	0.36
Total	100	Crude ash	6.57
		Acid insoluble ash	2.84

^1^ DM = dry matter. ^2^ One kilogram of premix contained the following: vitamin A 190,000 IU, vitamin D3 70,000 IU, vitamin E 2000 IU, Fe 2100 mg, Mn 3300 mg, Zn 3400 mg, Cu 460 mg, I 19 mg, Se 10 mg, Co 10 mg, Ca 130 g, P 30 g. ^3^ NE_mf_ = combined net energy. NE_mf_ was a calculated value, while the others were measured values.

**Table 2 animals-15-03123-t002:** Primer sequence used for RT-qPCR amplification.

Genes	Accession Number	Primer Name	Sequences of Primer (5′–3′)
Claudin-1	XM_027538401.1	Claudin-1-F	CTTCATCCTGGCGTTTCTGG
		Claudin-1-R	AGACTTTGCACTGGATCTGC
Occludin	XM_027520117.1	Occludin-F	AGATGCACGTTCGACCAATG
		Occludin-R	ATTACTCCGGGAGGAGAGGT
ZO-1	AJ313183.1	ZO-1-F	CAGCCAAAGAAGGCTTGGAG
		ZO-1-R	AGGTCAAGCAGGAAGAGGAC
β-Actin	XM_005887322.2	β-Actin-F	GCACCCAGCACAATGAAGAT
		β-Actin-R	CAGCTAACAGTCCGCCTAGA
*TJP1*	NM_001102482.1	TJP1-F	GGCAGCTTGAAAGACACGAT
		TJP1-R	GGCGGTTAAGTAGGACATGC
*PCNA*	NM_001034494.1	PCNA-F	CTTGGTGCAGCTAACCCTTC
		PCNA-R	TCCGCGTTATCTTCAGCTCT
*APAF1*	NM_001191507.1	APAF1-F	TTCATCCGCCAAGGTTCTCT
		APAF1-R	CTCCACCTGTCTTGAGCAGA
*FAS*	NM_174662.2	FAS-F	TCCTTCGTCAAAGACTGCCT
		FAS-R	CGCCATGACATCCTTGAACC
*MAP2K7*	XM_024994921.1	MAP2K7-F	CTGGGATTCTCCAGGCAAGA
		MAP2K7-R	TTGCTGAGTCGGACATGACT
*RASGRF*	XM_002689373.6	RASGRF-F	TTCATCCGCCAAGGTTCTCT CTCCACCTGTCTTGAGCAGA
		RASGRF-R	GGCAGCTTGAAAGACACGAT

**Table 3 animals-15-03123-t003:** Effects of different levels of PEO on the nutrient digestibility in heat-stressed beef cattle (%).

Items	Supplemental PEO ^1^ Level, mg/kg		SEM	*p*-Value
	0 (Control)	50		
Dry matter	87.353	88.357	1.539	0.550
Neutral detergent fiber	56.443 ^a^	60.160 ^b^	0.635	0.026
Acid detergent fiber	68.383	71.127	1.141	0.074
Ether extract	77.003	77.157	1.250	0.908
Calcium	48.953	50.000	1.856	0.603
Phosphorus	53.430	57.270	1.827	0.103
Crude protein	54.527 ^a^	61.033 ^b^	0.376	<0.001

^1^ PEO = *Pogostemon cablin* essential oil. ^a,b^ Means in a row not sharing a common letter are significantly different (*p* < 0.05).

**Table 4 animals-15-03123-t004:** Effects of PEO ^1^ on antioxidant parameters of ruminal tissues in heat-stressed beef cattle.

Items ^2^	Supplemental PEO Level, mg/kg		SEM	*p*-Value
	0 (Control)	50		
T-SOD, U/mL	11.33 ^b^	17.36 ^a^	0.913	0.001
GSH-Px, U/mL	15.83 ^b^	20.50 ^a^	1.387	0.015
T-AOC, U/mL	0.72 ^b^	1.02 ^a^	0.038	<0.001
MDA, nmol/mL	1.17 ^a^	0.40 ^b^	0.074	<0.001

^1^ PEO = *Pogostemon cablin* essential oil. ^2^ GSH-Px = glutathione peroxidase; T-AOC = total antioxidant capacity; T-SOD = total superoxide dismutase; MDA = malondialdehyde. ^a,b^ Means in a row not sharing a common letter are significantly different (*p* < 0.05).

**Table 5 animals-15-03123-t005:** Effects of PEO ^1^ on rumen fermentation and digestive enzyme activities in heat-stressed beef cattle.

Items ^2^	Supplemental PEO Level, mg/kg		SEM	*p*-Value
	0 (Control)	50		
Rumen fermentation				
pH	6.39	6.10	0.367	0.463
Ammonia-N, mg/dL	23.91 ^a^	13.72 ^b^	2.591	0.008
MCP, mg/mL	7.39	7.66	2.778	0.927
Acetate, mmol/L	36.64	40.97	12.105	0.733
Propionate, mmol/L	9.58 ^b^	12.69 ^a^	1.072	0.027
Butyrate, mmol/L	6.45	7.85	3.052	0.663
A:P	3.67	3.27	0.354	0.293
VFA, mmol/L	40.16 ^b^	59.35 ^a^	6.452	0.041
Digestive enzyme activities				
Lipase activity (U/L)	44.760	48.287	3.886	0.705
Cellulase activity (U/L)	50.037 ^b^	52.827 ^a^	0.859	0.031
α-amylase activity (U/L)	15.735	16.623	1.502	0.576

^1^ PEO = *Pogostemon cablin* essential oil. ^2^ MCP = microbial protein; VFA = volatile fatty acids; A:P = acetate to propionate ratio. ^a,b^ Means in a row not sharing a common letter are significantly different (*p* < 0.05).

**Table 6 animals-15-03123-t006:** Statistical table of KEGG pathway enrichment analysis of differential genes.

Pathway Name	Input Number	Background Number	*p*-Value
Protein processing in endoplasmic reticulum	22	210	7.92 × 10^−5^
Longevity regulating pathway–multiple species	9	65	1.79 × 10^−3^
Alzheimer disease	32	443	2.16 × 10^−3^
Inositol phosphate metabolism	9	78	5.37 × 10^−3^
Huntington disease	27	380	5.74 × 10^−3^
HIF-1 signaling pathway	12	130	8.75 × 10^−3^
Phagosome	16	218	2.22 × 10^−2^
Steroid biosynthesizer	4	21	1.05 × 10^−2^
Gap junction	10	103	1.14 × 10^−2^
PPAR signaling pathway	9	90	1.34 × 10^−2^
Fatty acid degradation	6	47	1.34 × 10^−2^
RNA degradation	9	92	1.53 × 10^−2^
Legionellosis	8	81	2.05 × 10^−2^
Tryptophan metabolism	6	52	2.14 × 10^−2^
Thermogenesis	22	330	2.29 × 10^−2^
AMPK signaling pathway	11	135	2.73 × 10^−2^
Insulin signaling pathway	12	155	3.08 × 10^−2^
Growth hormone synthesis, secretion and action	10	121	3.16 × 10^−2^
Valine, leucine and isoleucine degradation	6	57	3.20 × 10^−2^
Cysteine and methionine metabolism	10	110	3.98 × 10^−2^
Regulation of lipolysis in adipocytes	6	61	4.26 × 10^−2^
Apoptosis	15	218	4.27 × 10^−2^
Parathyroid hormone synthesis, secretion and action	9	113	4.88 × 10^−2^
Tight junction	14	205	4.96 × 10^−2^

**Table 7 animals-15-03123-t007:** Differentially expressed genes related to the epithelial barrier of the rumen.

Gene ID	Gene Name ^1^	Expression Quantity		Log2Fold Change		Expressing Trends
		Control	PEO ^2^ Group			
ENSBTAG00000011770	*TJP1*	10.80	14.37	1.04	4.52 × 10^−4^	Up
ENSBTAG00000000428	*BAK1*	16.04	28.13	0.81	9.52 × 10^−3^	Up
ENSBTAG00000006065	*PCNA*	22.99	35.12	1.49	3.58 × 10^−2^	Up
ENSBTAG00000012244	*TUBA1B*	226.84	423.89	1.28	2.19 × 10^−2^	Up
ENSBTAG00000004322	*FAS*	2.15	0.99	−0.99	2.16 × 10^−3^	Down
ENSBTAG00000021661	*APAF1*	1.27	0.59	−1.10	4.78 × 10^−2^	Down
ENSBTAG00000030259	*RASGRF2*	1.38	0.6	−2.05	3.95 × 10^−2^	Down
ENSBTAG00000010998	*CFLAR*	4.61	3.81	−0.77	1.78 × 10^−2^	Down
ENSBTAG00000050313	*BBC3*	1.99	1.28	−0.84	1.67 × 10^−2^	Down
ENSBTAG00000010639	*MAP2K7*	8.22	6.27	−0.39	1.64 × 10^−2^	Down

^1^ *TJP1* = Tight Junction Protein 1; *BAK1* = BCL2 Antagonist/Killer 1; *PCNA* = Proliferating Cell Nuclear Antigen; *TUBA1B* = Tubulin Alpha 1B; *FAS* = Fas Cell Surface Death Receptor; *APAF1* = Apoptotic Protease Activating Factor 1; *RASGRF2* = RAS Guanine Nucleotide-Releasing Factor 2; *CFLAR* = CASP8 and FADD-Like Apoptosis Regulator; *BBC3* = BCL2 Binding Component 3 (also known as PUMA, p53 Upregulated Modulator of Apoptosis); *MAP2K7* = Mitogen-Activated Protein Kinase Kinase 7. ^2^ PEO = *Pogostemon cablin* essential oil.

## Data Availability

The raw sequencing data used in this study were deposited in the NCBI Sequence Read Archive (SRA) under BioProject accession number PRJNA1310241. All data generated or analyzed during this study are included in this published article.

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
