# Peer review of "Impact of Dietary Supplementation with *Pogostemon cablin* Essential Oil on the Rumen Fermentation and Rumen Health in Heat-Stressed Beef Cattle"

_animals, 2025, doi:10.3390/ani15213123_

Round 1

Reviewer 1 Report

Comments and Suggestions for Authors

General Comments:

  • The experiment used a single concentration of PEO for supplementation; however, no data were provided regarding the rationale for the chosen concentration or any dose-response reference indicators. How can the authors confirm that the selected concentration is effective?
  • The experiment lasted for two months, but the initial body weights of the animals and changes in body weight during the study were not presented. These are critical indicators to assess the potential beneficial effects of PEO supplementation.
  • For the assay kits applied in the experiment, please provide the product model numbers and detailed sample pretreatment protocols. Currently, only the supplier names are listed.
  • Please recheck the number of significant figures used in the tables. Additionally, verify the capitalization and chemical formula accuracy of all terms presented in the tables.
  • Were all animals kept under high THI conditions throughout the entire experimental period? The authors should provide detailed temperature and humidity data to ensure that differences in feed intake or stress levels due to environmental variation did not confound the results.
  • Regarding the proposed mechanism for increased expression of tight junction proteins following PEO supplementation, please provide supporting literature references.

Specific Comments:

L97 Table 1

  • The feed composition analysis should include ash and AIA contents.
  • Please specify the calculation method used for NEmf.

L103

  • Please describe the methods and timing used for fecal sample collection.
  • What were the daily feeding amounts and feeding frequency during the trial? Was feed refusal recorded to calculate actual feed intake? Feed intake is a key indicator under heat stress conditions.

L106

  • If VFA and NH₃ were not analyzed in fecal samples, please clarify the purpose of adding sulfuric acid during sampling.

L118

  • Was fasting applied before slaughter? If so, for how long?
    Rumen content and fermentation characteristics can change significantly after fasting, which also alters enzyme and microbial activities.

L124

  • The cellulase described here should be clarified as CMCase, an endoglucanase. Please correct the EC number to EC 3.2.1.4.
  • Was the activity of exo-glucanase (EC 3.1.2.91) also analyzed? Effective fiber degradation in the rumen requires both types of enzymes.

L127

  • What sample preparation was conducted prior to VFA and NH₃ analysis in rumen fluid? Was centrifugation performed?

L137

  • Why was the dorsal sac selected as the sampling site?
  • Were samples taken from every animal? From only one site per rumen?
    In Figure 1, it states N = 50, but the total number of animals is 9 + 9 = 18. Please clarify the number of samples per group.

L233 Table 3

  • It is recommended to revise the title to: "...PEO on the nutrient digestibility (%)".
    Since the values already represent percentages, the "%" symbol does not need to be repeated in each item.
  • Please provide dry matter digestibility data.
  • CP digestibility appears to be listed twice; please check for duplication.

L263-264

  • Rumen fluid was filtered for enzyme activity assays. However, a substantial proportion of enzyme activity exists in the solid phase of rumen content. Please clarify how the analyzed enzyme activity represents the in vivo ruminal condition.

L266 Table 5

  • Please define the unit (U) used for enzyme activity measurements.
  • α-Amylase activity data appears duplicated; please verify.
  • Check the statistical significance markings for lipase activity for accuracy.

L367-368

  • Increased gastrointestinal motility may raise the dilution rate of the rumen and reduce feed retention time, potentially decreasing digestibility due to insufficient fermentation time.

L375-377

  • Please provide feed intake data to support the claim that PEO had a positive effect under heat stress conditions.

L395-397

  • Rumen papillae undergo keratinization and differ from intestinal villi in terms of absorptive properties. The cited reference [30] may not adequately explain these differences.

L409-411

  • If VFA production, microbial protein synthesis, and total protein digestibility increased, corresponding animal weight gain data should be presented to support these findings.

L439-441

  • Different essential oils may have distinct physiological effects and mechanisms of action. Please confirm that the essential oils used in the cited literature are the same or similar in composition and function to those used in this study.

L496-497

  • Which specific indicators were used to assess improved thermotolerance?
    Since the goal in beef production is weight gain and meat quality improvement, the biological impact of enhanced antioxidant activity or rumen barrier function should be demonstrated through tangible production outcomes.

Reviewer 2 Report

Comments and Suggestions for Authors

Interesting study.

Comments are in the attached document.

Comments on the Quality of English Language

There are few spelling and grammar errors. Some of them have been noted in the comment document.

Reviewer 3 Report

Comments and Suggestions for Authors

The manuscript addresses a relevant and timely topic, investigating the role of Pogostemon cablin essential oil (PEO) as a dietary additive to alleviate heat stress in beef cattle. Heat stress is a major concern in livestock production, and exploring natural feed additives for mitigation has both scientific and practical significance. The study integrates physiological, biochemical, and transcriptomic data, providing a multidimensional view of PEO effects. Overall, the manuscript is generally well-structured and presents interesting findings.

However, several points need to be addressed to improve:

  1. Novelty: Reference [13] (Chen et al., 2024, Animal Nutrition) already studied PEO in heat-stressed cattle. The authors present the work as novel, but the findings appear very similar to those reported by Chen et al. 2024. The manuscript demonstrates a high textual similarity of 35%, which raises concerns regarding originality.
  2.  Intake : The manuscript mentions reduced digestibility under heat stress but does not report actual intake data. Since feed intake is typically the first variable affected by heat stress, this omission represents a major gap.
  3. Production Parameters: No measurements of weight gain, feed efficiency, or carcass traits are reported. This omission limits the assessment of the economic or production relevance of PEO supplementation.
  4. Microbial Protein (MCP): The authors claim an increase in MCP, but Table 5 shows no significant difference. The discussion should be corrected.
  5. Slaughter and Sampling Conditions:
    • Animals were slaughtered 500 m from the barn, but no details are provided on pre-slaughter fasting or stress handling, which can influence rumen parameters.
    • There is no mention of the time elapsed after feeding before slaughter, which strongly affects fermentation measurements.
    • The number of rumen papillae measured per animal is not specified.
  6. Digestibility Data: In line 228, the text states: “Table 3 shows that supplementing with 50 mg/kg PEO increased the CP (P <0.001) and NDF (P = 0.026) digestibility, and tended to increase the NDF (P = 0.074) digestibility.” The second mention of NDF is likely a typographical error and should refer to ADF digestibility.
  7. RNA-seq Analysis: Only three animals per group were included for RNA-seq, which is insufficient for analysis.
  8. Rumen Morphology: Figures should include scale bars, and image quality should be improved.
  9. Fecal Sampling: Fecal samples were pooled by group, which prevents true individual-animal statistical analysis of digestibility.
  10. Discussion: Comparison with studies on other essential oils in cattle  is recommended to contextualize the findings.

Round 2

Reviewer 1 Report

Comments and Suggestions for Authors

The revised manuscript had many improvements in the discussion section. However, there were still some typing errors in the Tables.

Table 1:

Nacl →NaCl

NaHCO3 → NaHCO3

DM→ DM1

Premix → Premix2

NEmf/(MJ/kg) → NEmf/(MJ/kg)3

Table 4.

Effects of PEO on … → PEO1

Items → Items2

Table 5, Table 7 :

Showed the same superscript error as Table 4

Author Response

Comments 1: The revised manuscript had many improvements in the discussion section. However, there were still some typing errors in the Tables.

Table 1:

Nacl →NaCl

NaHCO3 → NaHCO3

DM→ DM1

Premix → Premix2

NEmf/(MJ/kg) → NEmf/(MJ/kg)3

Table 4.

Effects of PEO on … → PEO1

Items → Items2

Table 5, Table 7 :

Showed the same superscript error as Table 4

Response 1: We are grateful to the reviewer for their meticulous review and for identifying the typographical errors present in the tables. These errors have now been corrected, and all amendments are highlighted in yellow in the revised manuscript for easy identification.

Reviewer 3 Report

Comments and Suggestions for Authors

Accept in present form

Author Response

Comments 1: Accept in present form

Response 1: We are deeply grateful to the reviewer for their time and for the positive decision of "Accept in present form." We sincerely appreciate their insightful comments and suggestions throughout the review process, which have significantly improved the quality of our manuscript.